# Latest Contributions of Genomics to T-Cell Acute Lymphoblastic Leukemia (T-ALL)

**DOI:** 10.3390/cancers14102474

**Published:** 2022-05-17

**Authors:** Eulàlia Genescà, Celia González-Gil

**Affiliations:** Institut d’Investigació Contra la Leucemia Josep Carreras (IJC), Campus ICO-Germans Trias i Pujol, Universitat Autònoma de Barcelona, 08916 Badalona, Spain; cgonzalez@carrerasresearch.org

**Keywords:** T-cell acute lymphoblastic leukemia, genomics, non-coding, germline, aging, relapse

## Abstract

**Simple Summary:**

Thanks to the use of high-resolution genetic techniques to detect cryptic aberrations present in T-ALL, we now have a clearer view of the genetic landscape that explains the particular oncogenetic processes taking place in each T-ALL. We also have begun to understand relapse-specific mechanisms. This review aims to summarize the latest advances in our knowledge of the genome in T-ALL and highlight the areas where the research on this ALL subtype is progressing, thereby identifying the key issues that need to be addressed in the medium-to-long term to move forward in the applicability of this knowledge into clinics.

**Abstract:**

As for many neoplasms, initial genetic data about T-cell acute lymphoblastic leukemia (T-ALL) came from the application of cytogenetics. This information helped identify some recurrent chromosomal alterations in T-ALL at the time of diagnosis, although it was difficult to determine their prognostic impact because of their low incidence in the specific T-ALL cohort analyzed. Genetic knowledge accumulated rapidly following the application of genomic techniques, drawing attention to the importance of using high-resolution genetic techniques to detect cryptic aberrations present in T-ALL, which are not usually detected by cytogenetics. We now have a clearer appreciation of the genetic landscape of the different T-ALL subtypes at diagnosis, explaining the particular oncogenetic processes taking place in each T-ALL, and we have begun to understand relapse-specific mechanisms. This review aims to summarize the latest advances in our knowledge of the genome in T-ALL. We highlight areas where the research in this subtype of ALL is progressing with the aim of identifying key questions that need to be answered in the medium-long term if this knowledge is to be applied in clinics.

## 1. Introduction

Acute Lymphoblastic Leukemia (ALL) is an infrequent aggressive cancer with an age-standardized incidence of 0.68/100,000 individuals that is most common in children under 5 years of age [1]. ALL includes B- and T-cell lineage subtypes. While these diseases share many similarities, such as some general cancer genetic lesions in cell cycle regulators such as *CDKN2A/B* and *MYC* [2], their genetic background, origin, and outcome are distinct. The T-ALL subtype accounts for 10–15% of pediatric and 25% of adult ALL cases. It is more frequent in males than in females, and more often presents at younger ages, being considered an AYA (adolescent and young adult)-affecting disease [3]. Although intensive chemotherapy treatments have been crucial for increasing the survival rate of affected children up to 93% [4], globally, adult survival rates are still very poor, remaining below 50% [5]. The main cause of treatment failure and worse outcome is relapse [6,7]. In this scenario, T-ALL patients with primary-resistant or relapsed leukemia present a dismal prognosis, mainly due to the rapid progression of the recurrent course of the disease and the lack of effective therapeutic options [7,8].

Initial genetic studies based on the use of cytogenetics and FISH helped identify some recurrent chromosomal alterations in T-ALL at the time of diagnosis, but it proved difficult to determine their prognostic impact because of their low incidence in the specific T-ALL cohort analyzed. Genetic knowledge flourished with the application of genomic techniques, first with the analysis of gene expression profiles (GEPs), followed by the application of comparative genomic hybridization arrays (CGHas), and later, the next generation sequencing (NGS) technique. Consequently, we now have a clearer appreciation of the different T-ALL genetic subtypes at the time of diagnosis and are beginning to understand relapse-specific mechanisms.

The aim of this review is to summarize the latest scientific advances in our knowledge of the T-ALL genome. We will highlight the areas where the research on this ALL subtype is progressing, thereby identifying the key issues that need to be addressed in the medium-to-long term to move forward in the applicability of this knowledge into clinics.

## 2. T-ALL Classification by Differentiation Stage

The main contribution of genomic techniques in T-ALL has been to show that the blockade of the differentiation process that occurs in the lymphocyte is the consequence of specific genetic alterations occurring in pre-leukemic stages. From a historical perspective, the first studies to classify T-ALL aroused with the use of the gene expression array (GEa). This revealed that structural abnormalities, mainly rearrangements identified in T-ALL by karyotyping as a rare event [9,10,11,12,13,14,15,16,17,18,19], led to much overexpression of (1) basic helix-loop-helix (bHLH) transcription factor genes such as *TAL1*, *TAL2*, and *LYL1*; (2) LIM-only (*LMO*) domain genes such as *LMO1* and *LMO2*; and (3) homeobox genes such as *TLX1* (*HOX11*) and *TLX3* (*HOX11L2*). Supervised analysis of the expression data revealed a correlation between the expression of these transcription factors (TFs) and a lymphocyte-specific differentiation arrest time point. Three main groups (clusters) were obtained: (1) immature, (2) HOXA, and (3) TAL [20,21]. To describe this in more detail, HOXA samples cluster with the Pre-T1 (CD34 + CD1a+) and Pre-T2/Pre-T3 (CD4+, CD8αβ-, CD3- and TCR αβ-) sub-populations, together with the *TLX1*-expressing cases and some cases expressing *TLX3*. TAL samples cluster with subpopulations corresponding to thymocytes with a pre-TCR (Beta selection). Of note, samples from the immature group included some TLX3-expressing leukemic samples and clustered with the most immature early T-cell precursor and pro-T subpopulations [21]. Later on, the addition of copy number data, generated by CGHa, together with the use of NGS, yielded a clearer view of the genetic determinants that define these groups. Thus, according to the differentiation arrest time point of the blast cell we can differentiate the immature subtype, characterized by the absence of CD8 and CD1a immunomarkers, high levels of expression of *LYL1* [20] and *MEF2C* [22] TFs, and the absence of bi-allelic deletions in TCRγ [23]. Within the immature T-ALL leukemias, the early T-cell precursor ALL (ETP-ALL)—defined by the absence of CD1a and CD8 immunomarkers and the presence of stem cells or myeloid markers such as CD117, CD34, HLA-DR, CD13, CD33, CD11b, and CD65 [24], together with negative or dim CD5 expression, defined as expression in <75% of the blasts; rarely presents *CDKN2A/B* deletions and *NOTCH1/FBXW7* mutations [25,26,27]. In addition, mutations affecting epigenetic regulators and transcription factors governing hematopoietic and T-cell development are also frequently observed in this immature subtype [25,28,29,30,31]. We will discuss this subgroup in detail later. The cortical subtype is characterized by the expression of CD1a, and often both CD4 and CD8 immunomarkers are also found. At the genetic level, the group is characterized by aberrant expression of *TLX1*, *TLX3*, and *HOXA* genes (*HOXA5*, *HOXA9*, *HOXA10*) [20,21] and the overexpression of the *NKX2-1* rearranged gene [22]. Other specific alterations in this subset of T-ALL have been found in *PHF6*, *DNM2*, *BCL11B*, *CDKN1B*, and *RB1* [32]. The mature subtype is characterized by blasts expressing CD4 or CD8 and surface CD3 immunomarkers [20] and the presence of Tαβ cell-receptor rearrangements [21]. The genetic hallmark of the group is the activation of the *TAL* oncogene [20], together with the presence of del(6)(q) [33] and mutations in *PIK3R* and *PTEN* [32]. Figure 1 summarizes these findings.

Collectively, the data generated during twenty years of genomics research into T-ALL highlight the importance of using high-resolution genetic techniques. These not only make it possible to detect the cryptic aberrations present in T-ALL but also to define primary genetic events that determine the acquisition of the secondary genetic events necessary for transforming T-cell progenitors. This thereby explains the particular oncogenetic processes taking place in each T-ALL. We are moving towards an immuno-genetic T-ALL classification. One of the knowledge gaps here relates to T-ALL leukemias with unidentified primary events, although the driving TF is aberrantly expressed. In those cases, it is reasonable to argue that alterations in regulatory regions and/or in enhancers of the TF can alter its expression.

### 2.1. Non-Coding Mutations

Analysis of non-coding data generated by whole genome sequencing (WGS) or direct target sequencing is an emerging area of investigation, but recently published data have shown that small (or not) insertions and deletions generating novel regulatory sequences can explain the overexpression seen in TFs, such as *TAL1* and *LMO1*, that have no detectable primary rearrangements. Here we provide a summary of the non-coding alterations identified in *TAL1*, *LMO1/2*, and other important T-ALL oncogenes such as *MYC* and *PTEN*. Non-coding variants discovered in T-ALL are also summarized in Table 1.

The first non-coding alterations found to affect T-ALL corresponded those affecting *MYC* TF activation by *NOTCH1* [34]. Recurrent focal duplications at chromosome 8q24 were identified in a CGH screening in 8/160 (5%) of the T-ALL cases analyzed. The amplification was located +1427 kb downstream of *MYC*. The alteration was named N-Me for NOTCH *MYC* enhancer. The region was shown to interact with the *MYC* proximal promoter and induced orientation-independent *MYC* expression in reporter assays. Analysis of N-Me knockout mice demonstrated a selective role of this regulatory element in the development T-ALL in a NOTCH1-induced T-ALL model [34].

Concomitant to this finding, a non-coding alteration located in a regulatory region of *TAL1* was also identified. A 12-bp indel that introduced two consecutive de novo binding motifs for the MYB TF, creating a super-enhancer, was identified using chromatin immunoprecipitation experiments and sequencing (CHIP-seq) in the Jurkat cell line [35]. The screening of 146 unselected pediatric primary T-ALL samples collected at diagnosis revealed that eight cases (5.5%) contained the 2–18-bp heterozygous indel, confirming the in vitro results. Sequencing DNA from remission bone marrow samples in two available cases showed wild-type sequences at this site, indicating that the mutations were somatically acquired in the blasts. The authors suggested that the initial mechanistic event in the aberrant super-enhancer formation was the recruitment of CBP by MYB, followed by abundant H3K27 acetylation, which facilitated the binding of a core complex composed of RUNX1, GATA-3, and *TAL1* itself [35]. These results were validated in a study that set out to identify non-coding mutations in 31 pediatric T-ALL cases from the available WGS data. Non-coding *TAL1* mutations were significantly associated with *TAL1* expression, implying a cis-regulating effect. Consistent with previous results, a similar, MYB binding-dependent *TAL1* promoter activation mechanism was described [36].

Subsequent studies used the same experimental procedure; (1) identification of a core sequence by CHIP-seq in T-ALL cell lines; (2) identification of epigenetic marks supporting transcriptional activity in the region and the transcriptional complex; and (3) validation of the alteration in a large cohort of pediatric T-ALL cases revealed a C-to-T single nucleotide transition occurring as a somatic mutation in the non-coding sequence 4 kb upstream of the transcriptional start site of the *LMO1* TF. This single nucleotide alteration gives rise to an APOBEC-like cytidine deaminase mutational signature and generates a new binding site for the MYB transcription factor, leading to the formation of an aberrant transcriptional enhancer complex that drives high levels of expression of the *LMO1* oncogene, similar to the *TAL1* super-enhancer. Sequencing 187 pediatric primary T-ALL samples collected at diagnosis identified four patients (2.14%) with the same heterozygous mutation [37]. As with the *TAL1* indel, the *LMO1* aberrant MYB-dependent super-enhancer was confirmed by the WGS study [36]. However, Shaoyan and colleagues also identified an intrachromosomal inversion event that juxtaposed the active promoter of the *MED17* gene with the coding sequence of *LMO1*, leading to the expression of *LMO1*. This highlights how other structural abnormalities may help explain the abnormal expression of this TF [36]. In the case of *LMO2*, a heterozygous 20-bp duplication in PF-382 cells and a heterozygous 1-bp deletion in DU.528 cells were identified in the same way, both being located closer to a region recently described as an intermediate promoter. These alterations were not limited to T-ALL cell lines, since heterozygous mutations in the *LMO2* intron 1 were detected in diagnostic samples from 6/160 of the pediatric and 9/163 of the adult T-ALL cases sequenced [38]. The confirmatory WGS study, in this case, confirmed the non-coding mutations in regulatory regions of *LMO2*. However, in this case, they were not associated with *LMO2* gene expression [36] (Table 1).

An intronic sequence of 550-kb situated in the neighborhood of the *RNLS* gene and downstream of the *PTEN* gene has very recently been identified and found to interact strongly with the *PTEN* promoter. The presence of enhancer marks in this region, including high levels of H3K27ac and H3K4me1, together with binding of CTCF, BRD4, and ZNF143, defined this region as a *PTEN* enhancer (PE). Screening for genetic lesions in this region in human primary samples identified five cases (5/398, 1.25%) with focal deletions encompassing PE. The deletion was homozygous in two of these samples, and additional simultaneous deletions targeting the coding region of *PTEN* were observed in four of the five cases. Analyses of 1415 BCP-ALL samples failed to identify the same deletion, showing that these alterations are restricted to the T-ALL subtype [39] (Table 1).

Identification of non-coding variants is a burgeoning area of research in which much information is yet to be gathered. The contribution of non-coding sequences to oncogenetics remains largely unknown. In addition to alterations in promoters, regulatory regions and enhancers, as well as other non-coding regions such as intergenic and splicing site sequences, will help refine the immuno-genetic T-ALL classification.

### 2.2. T-ALL Related Immature Subtypes

Application of WGS and whole transcriptome sequencing has also served to better characterize rare subtypes such as the immature T-ALL leukemias and to provide insight into the cell of origin of these subtypes. This group includes T/Myeloid mixed phenotype acute leukemias (T/M MPALs) and ETP-ALL, which are characterized by different combinations of myeloid and T-lymphoid antigen expression [40]. Other immunophenotypically identified immature T-ALL subtypes include the pro-T [40] and the near-ETP [32,41] forms, but we do not know their genetic basis or clinical implications. Thus, childhood ETP-ALL presents cytokine-activating somatic mutations and mutations in genes involved in the RAS signaling pathway (e.g., *NRAS*, *KRAS*, *FLT3*, *IL7R*, *JAK3*, *JAK1*, *SH2B3*, and *BRAF*), genetic alterations that inactivate genes involved in hematopoietic development (e.g., *GATA3*, *ETV6*, *RUNX1*, *IKZF1*, and *EP300*), and mutations in histone modifier genes (e.g., *EZH2*, *EED*, *SUZ12*, *SETD2*, and *EP300*). It is of note that the mutational spectrum identified in ETP leukemia was similar to that of acute myeloid leukemia (AML) with poor prognosis, in which affected pluripotent genes lend this subtype a myeloid-like profile [29]. In the case of adult ETP-ALL, exclusively genetic alterations have been detected in the *DNMT3A* gene (frequency range from 12 to 16%) [30,31,42,43], in addition to the aforementioned mutations. Specifically, *DNMT3A* mutations are associated with patients aged >60 years with ETP-ALL features [43]. *FAT1* (25%, 17/68) and *FAT3* (20%, 14/68) cadherins are other mutations exclusively found in adult ETP-ALL [30] (Figure 2). In addition to point mutations, structural abnormalities such as rearrangements affecting *KMT2A*, *MLLT10*, *NUP214*, or *NUP98*, which trans activate *HOXA* genes, are often detected in ETP-ALL cases [44,45]. Overexpression of the *BCL11B* gene due to different structural abnormalities including translocations (i.e., t(2;14)(q22.3;q32), t(6;14)(q25.3;q32), hijacking super-enhancers, and other fusion genes has been very recently described as present in one third of ETP-ALL and T/myeloid mixed phenotype acute leukemia (T/M MPAL) cases with a very distinct expression profile [46,47].

In the case of T/M MPAL leukemias, a study cohort including 49 pediatric cases of this mixed phenotype showed a high number of copy number alterations (CNAs) (average of 4.5 (0–35)) in this subtype compared to KMT2Ar MPAL leukemias. Alterations in genes encoding transcriptional regulators were also detected in the T/M MPAL cases (i.e., WT1, *ETV6*, *RUNX1*, and *CEBPA*) [48]. Alterations in JAK–STAT signaling were also common in these leukemias together with mutations in genes encoding epigenetic regulators (69% of cases), including inactivating mutations in *EZH2* (16%) and *PHF6* (16%). Analysis of the transcriptome sequencing identified chimeric in-frame fusions in 15/40 cases, including *ZEB2*–*BCL11B* (*n* = 3) and several fusions involving the *ETV6* gene [48].

Comparison of the genetic profiles of ETP-ALL and T/M MPAL with non-immature T-ALL leukemias in pediatric cases has shown that the core TF driving T-ALL (*TAL1*, *TAL2*, *TLX1*, *TLX3*, *LMO1*, *LMO2*, *NKX2- 1*, *HOXA10*, and *LYL1*) is less frequently altered in T/M MPAL and ETP-ALL. Other alterations that are common in T-ALL, such as *MYB* amplification, *LEF1* deletion, and *CDKN2A/B* deletions, are also rare in both types of immature leukemia. By contrast, *WT1* alterations are common in T/M MPAL and ETP-ALL, but not in non-immature T-ALL [48]. A similar analysis conducted in adult MPAL leukemias observed that, while myeloid-T/M MPAL and T-ALL shared a number of mutations in common, there were also differences. *PHF6* and *JAK3* mutations, each detected in 21.4% of the adult T-ALL cases analyzed, were not detected in myeloid-T/M MPAL. In contrast, *ASXL1* (11.1%) and *FLT3* (11.1%) mutations were detected in T/M MPAL but not in T-ALL [49]. Collectively, these observations imply that different primary events can drive specific T-ALL subtypes. A summary of these data can be seen in Figure 2.

## 3. (Epi)genetic Modification

The systematic screening of T-ALL genomes has revealed T-ALL as one of the tumors with the highest frequency of mutations in genes that encode proteins involved in epigenetic regulation [50]. Hence, the field of epigenetics, particularly DNA methylation, is currently being extensively explored in the search for specific methylation patterns that help to explain the oncogenic evolution of pre-leukemic T-cells; to identify specific de-regulated genes to use as a prognosis marker; and to delineate new therapy strategies using DNA methylation inhibitors (iDNMTs) such as 5-azacitidine (vidaza, AZA) and 5-aza-20 -deoxycytidine (decitabine, DAC) [51].

Initial epigenetic studies were focused on determining the methylation status of the promoter of specific genes playing a role in the T-ALL oncogenic process such as *CDKN2A/B*. It was observed in T-ALL patients that the percentage of promoter methylation in the *CDKN2B* and *CDKN2A* genes ranged between 46% and 68% and between 0% and 12%, respectively, in pediatric cohorts ([52,53,54]. In the case of adult T-ALL cohorts, the percentage of *CDKN2B* gene promoter methylation varied from 16% to 49% and was 1% for the *CDKN2A* promoter [55,56,57,58,59]. In T-ALL, the *CDKN2B* methylation status was associated with an immature immunophenotype [58] and with ETP-ALL features [59]. Further investigation of the methylation status in cancer cells using wide genomic approaches (i.e., methylation arrays) have observed that generally malignant cells display a DNA hypermethylation pattern at specific CpG islands; globally, this observation is called a CpG island methylator phenotype (CIMP). CIMP+ in T-ALLs has been associated with a better EF and OS as compared to CIMP− leukemias [60]. Notably, these findings have been confirmed in both pediatric [61] and adult [62] T-ALL cohorts, reinforcing the idea that aberrant DNA methylation might act as a clinically relevant biomarker in human T-ALL. Comparison of the global methylation profile of T-ALL samples with that of normal thymocytes observed that the methylation profile of CIMP− cases was close to normal CD3+ and CD34+ thymocytes [60,61]. That was interpreted as an indication of a shorter proliferation history of the CIMP− blasts as compared to CIMP+ cases [63]. Together, these findings indicate that CIMP− cases are characterized by a hypomethylation pattern that results in a young mitotic age and shorter proliferative history of leukemic cells; at the same time, however, it might be considered as a marker of higher aggressiveness of leukemic cells. In CIMP+ cases, the disease latency is longer, which is reflected by higher methylation acquired during the aging of pre-leukemic cells, [61,63,64]. Altogether, these results indicated that aberrant methylation is likely not a driving force of T-ALL onset and progression but is rather related to the proliferative history of the cells. These concepts have been nicely reviewed by Natalia Mackowska et al. [65] in a recent publication.

## 4. Germline Variants and Predisposition Alleles

As already mentioned, the incidence of ALL is higher in children between 2 and 5 years than in adults. This, together with the fact that the onset of ALL in children is shorter than in adults, supports the hypothesis of an inherited genetic basis for ALL susceptibility. Here, genetic and non-genetic determinants (i.e., environmental factors) could help explain the etiology of the disease in the very young. A focus on genetic determinants and the investigation and identification of germline variants in “presumably” sporadic cancers such as ALL has rapidly become more common in recent years. It is important to comment, however, that this new area of investigation has some limitations, since predisposition variants are much rarer in hematological cancers such as ALL than in pediatric solid tumors [66], and the incidence of T-ALL in pediatric cases corresponds to 10–15% of the global childhood ALL. Nonetheless, some germline variants have been identified, with possible involvement in disease predisposition. Table 2 summarizes these germline variants. Essentially two types of germline variants can be identified, according to the technical approach used to detect them. Thus, classical genome wide association studies (GWAS) identify predisposition SNPs (alleles) often observed to be altered in T-ALL patients compared with normal (disease-free) cohorts. On the other hand, sequencing of germline DNA using WGS or TDS approximations identifies pathogenic variants in the germline DNA of T-ALL patients with an implication in the disease. The main germline types may be also distinguished by their functional involvement: those contributing to the development of the T-ALL and those affecting the response to specific drugs used during T-ALL treatment (pharmacogenomics). We consider all types below. Nonetheless, germline variants in adults should also be considered in sporadic cancers since common variants can only explain a limited percentage of the genetic burden in cancer [67], and classic tumor suppressor genes (i.e., *TP53*) have been found to harbor rare, predisposing alleles [68]. In this context, a recent publication analyzed the landscape of pathogenic variants in the germline DNA of 10,389 individuals with different types of cancer. Strikingly, 8% of cases were found to carry confirmed or probable pathogenic germline variants, ranging in prevalence from 22.9% in pheochromocytoma and paraganglioma to 2.2% in cholangiocarcinoma [69]. These results dispel the long-held view that germline variants are “non-relevant” in the context of sporadic cancers, even in adult cancers, and will certainly have ramifications in the clinical context.

### 4.1. Germline Predispostion Alterations Contributing to the Development of T-ALL


*USP7*


A GWAS of 1191 children with T-ALL and 12,178 control subjects identified eight *USP7* SNPs of genome-wide significance, four of which (rs61426394, rs74010349, rs59591814, and rs74010351) clustered close to the *USP7* transcription start site in a region marked by the strong promoter-active histone modifications H3K4me3 and H3K27ac in two of the T-ALL cell lines analyzed. The region also overlapped with multiple open chromatin segments, suggesting a direct influence of the SNPs on *USP7* transcription. The rs74010351 variant exhibited the most statistically significant difference in a reporter gene transcription assay between the reference and risk alleles. Functional assays suggested that this T-ALL risk allele was located in a putative cis-regulatory DNA element, where it had negative effects on *USP7* transcription. The *USP7* risk allele was specifically restricted to the T-ALL subtype of ALL and was overrepresented in individuals of African descent, thereby contributing to the higher incidence of T-ALL in this ethnic group [70].


*ATM*


The major feature of ataxia-telangiectasia (A-T) is the increased risk of cancer, particularly of lymphoid malignancies. An interesting risk locus was identified when testing *ATM* (A-T-mutated gene) involvement in leukemia in 39 pediatric T-ALL patients from Israel. Two types of sequence changes —truncating and missense mutations— were identified in eight T-ALL germline samples. It was of particular interest that these eight patients also presented somatic *ATM* mutations. To assess the frequency of A-T heterozygote carriers in childhood T-cell ALL Israeli patients, a control population matched for ethnicity (North African origin (NAJs): *n* = 12, T-ALL vs. 100 non-T-ALL; Arab: *n* = 14, T-ALL vs. 100 non-T-ALL) association study was conducted. This revealed a 12.9-fold higher incidence of A-T carriers in the T-ALL population than in the normal controls, indicating an association between the A-T carrier and T-ALL (*p* = 0.004), and consequently with the susceptibility to develop a T-ALL. The results also support the model of predisposition to cancer in A-T heterozygotes [71].


*IKAROS*


Exploring the germline DNA of a 10-year-old child with a T-ALL developed 10 years after the diagnosis of a primary combined immunodeficiency, a novel heterozygous *IKZF1* mutation, c.476A > G (p.N159S), was identified in all specimens obtained from the patient. The p.N159S mutation was located within the second N-terminal zinc finger of IKAROS encoded by exon 4, which is essential for its localization to pericentromeric heterochromatin (PC-HC) and its DNA-binding function. The mutant protein displayed diffuse nuclear staining, suggesting that the correct recognition of target DNA sequences by IKAROS was impaired. In vivo experiments performed in immunodeficient mice demonstrated the role of this germline mutation in cooperating with the activation of the *NOTCH1* signaling pathway to develop the disease [72].


*RUNX1*


Initial studies in the late nineties demonstrated that germline mutations in *RUNX1* define a familial platelet disorder with predisposition to myeloid malignancy (FDP-MM). FDP-MM is an autosomal dominant disorder characterized by variable penetrance of quantitative and/or qualitative platelet defects with a tendency to develop hematological malignancies. The overall lifetime risk of progression to a myeloid disease (or rarely T-ALL) is 44% [77] and most commonly occurs in adulthood [78]. A recent study, that includes the largest series of familial FDP-MM (35 families), has provided a comprehensive overview of all genetic mutations and associated disease phenotypes. Here, two new *RUNX1* germline variants (36171607G > A and 36231773 C > T) were correlated with a high risk of developing T-ALL [73]. The influence of *RUNX1* on predisposing patients to ALL was subsequently assessed by sequencing germline DNA in 4836 children with BCP-ALL and 1354 with T-ALL. Thirty-one and 18 germline *RUNX1* variants were detected, respectively. Of these, *RUNX1* variants in B-ALL consistently showed minimal damaging effects, whereas six T-ALL-related variants (p.K117*, p.A142fs, p.S213fs, p.R233fs, p.Y287*, and p.G365R) caused a significant reduction in *RUNX1* activity as a transcription activator *in vitro*. Further, WGS identified a *JAK3* mutation as being the most frequent somatic genomic abnormality in T-ALL with germline *RUNX1* variants. Co-expression of the *RUNX1* variant and *JAK3* mutation in hematopoietic stem and progenitor cells in mice gave rise to T-ALL with the ETP-ALL phenotype, confirming the highly deleterious role of *RUNX1* germline variants in T-ALL [74].

### 4.2. Predisposition Alleles Affecting Drug Response


*NT5C2*


*NT5C2* is a ubiquitously expressed cytosolic nucleosidase in charge of maintaining intracellular nucleotide pool homeostasis by promoting the clearance of excess purine nucleotides from cells [79,80]. *NT5C2* preferentially dephosphorylates the 6-hydroxypurine monophosphates inosine monophosphate (IMP), guanosine monophosphate (GMP), and xanthosine monophosphate (XMP), as well as the deoxyribose forms of IMP and GMP (dIMP and dGMP), facilitating the export of purine nucleosides. In the context of ALL therapy, activating mutations in the *NT5C2* gene accelerates the dephosphorylation and inactivation of 6-MP intermediate metabolites and thereby reduces the amount of deoxy thioguanosine triphosphate (TdGTP) available for DNA incorporation during the treatment, which in turn compromises the cytotoxicity of this drug [81].

With the aim of identifying germline DNA variants associated with 6MP pharmacokinetics, a GWAS study of 1009 patients undergoing maintenance therapy for ALL was undertaken. The propensity for DNA-TG incorporation in the discovery cohort (454 patients) was significantly associated with three intronic SNPs in *NT5C2*. Only one of these (rs72846714 at 10q24.3) was significantly associated with DNA-TG incorporation at any treatment step. The association was confirmed in a validation cohort (555 patients). Targeted sequencing of the *NT5C2* gene did not identify any missense variants associated with the SNP. However, the rs72846714 was associated with a more frequent occurrence of relapse-specific *NT5C2* gain-of-function mutations, implying cooperation between gain-of-function *NT5C2* mutations and the germline SNP in relapse [75]. The association between rs72846714 SNP and *NTC5C2* expression was subsequently confirmed [76]. The rs72846714 SNP was not located in a regulatory element of the *NT5C2*. Instead, its association signal was explained by linkage disequilibrium with a proximal functional variant rs12256506, which, in this case, activates *NT5C2* transcription in *cis*. The same study identified another SNP (rs58700372) that directly alters the activity of an intronic enhancer (transactivation), whose variant allele is linked to higher transcription activity of the nucleosidase and therefore to reduced 6-MP metabolism [76].

## 5. Clonal Hematopoiesis and Aging

Some data suggest that a person typically acquires 10 to 20 non-pathogenic passenger mutations per stem cell by middle age [82], and normal hematopoietic stem cells (HSCs) may acquire approximately 0–1 exonic mutation per decade of life [83]. It is therefore not surprising that HSC populations of elderly humans are somatic mosaics. Some of these mutations (driver mutations*) may confer a survival advantage over non-mutated stem cells, resulting in proliferation of a clonal population or clonal hematopoiesis (CH). A large study of an Icelandic population found that CH was present in 0.5% of those younger than 35 years, but in more than 50% of subjects who were older than 85 years of age [84], arguing in favor of an increased pool of mutational events driving CH in the elderly.

In their review, Silvera and Jaiswal [85] employed two terms to define individuals who showed evidence of CH but lacked signs of a current or previous hematological malignancy. The first is clonal hematopoiesis of indeterminate potential (CHIP) [86]. CHIP defines a state of CH that arises from a somatic mutation affecting a particular set of genes known to be drivers of hematological malignancy: *DNMT3A*, *TET2*, *JAK2*, *SF3B1*, *ASXL1*, *TP53*, *CBL*, *GNB1*, *BCOR*, *U2AF1*, *CREBBP*, *CUX1*, *SRSF2*, *MLL2*, *SETD2*, *SETDB1*, *GNAS*, *PPM1D*, and *BCORL1* [86]. To be categorized as CHIP, a detected somatic variant must be present at a variant allele frequency (VAF) of at least 2% [85]. The second term is age-related clonal hematopoiesis (ARCH) [87]. The difference between this and CHIP is that a recurring alteration is identified as the possible cause of ARCH. Potential copy-number variants, together with mutations in candidate driver genes and epigenetic drift, have to be considered in ARCH development [88]. Despite the name and classification, its relevance is that any form of CH has the potential to transform [85,89] in myeloid [90] or even in lymphoid [89] neoplasms. Therefore, these mutations must be considered in a clinical context.

In this regard, the mutations found in the *DNMT3A* in adult T-ALL are of particular interest. As we have explained, mutations in this gene have been exclusively found in adult T-ALL and are associated with patients aged >60 years [43] as well as with immature subtypes such as the ETP-ALL [30,31,42,43]. Analysis of leukemic and non-leukemic cells of adult T-ALL patients showed that *DNMT3A* mutations were present in the non-leukemic fraction in two of the eight patients analyzed, suggesting that immature T-ALL cases could be derived from a CHIP event [42]. Similar results were obtained in an extended cohort, in which *DNM3TA* mutations were detected in polymorphonuclear cells, as a source of non-leukemic cells, and in leukemic cells [43]. VAFs in the *DNMT3A* gene were lower than or the same as those in the leukemic cells of all seven patients studied, suggesting a role for the *DNMT3A* mutations as a founding event in the development of T-ALL in the elderly and implying that the mutations could arise in an uncommitted myeloid-lymphoid progenitor [43]. In both studies, however, the possible germline origin of *DNMT3A* mutations could not be ruled out. More importantly, adult T-ALL patients with *DNMT3A* mutations have been significantly associated with worse clinical outcomes, a higher cumulative incidence of relapse (HR 2.33, 95% CI: 1.05–5.16, *p* = 0.037), and markedly poorer event-free survival (HR 3.22, 95% CI: 1.81–5.72, *p* < 0.001) and overall survival (HR 2.91, 95% CI: 1.56–5.43, *p* = 0.001) [42].

A driver mutation is an alteration that gives a cancer cell fundamental growth advantage for its neoplastic transformation. It differs from passenger mutations in that these do not necessarily determine the development of the cancer.

## 6. Relapse

Although intensive chemotherapy treatments have been crucial for improving survival to up to 93% of T-ALL-affected children, the survival rates in adults are still very poor, remaining below 50% [5]. Under both scenarios, T-ALL patients with primary-resistant or relapsed leukemia have a dismal prognosis, mainly due to the rapid progression of the recurrent course of the disease and the lack of therapeutic options. Application of multi-omics techniques in matched-diagnosis (DX), germline, or remission and relapse (RE) DNA samples to identify private and shared variants at DX and RE have helped us understand relapse mechanisms and identify target drugs to apply as new therapies.

The largest cohort studied until now, employing triplets, assessed the mutational profile, including CNVs, of 175 ALL patients (*n* = 149 pediatric and *n* = 26 adult cases; 129 BCP vs. 46 T-ALL) [91]. The WGS revealed that 52% of the relapse samples contained most of the genetic lesions present in the major clone at diagnosis. Overall, there were significantly more relapse-specific mutations than diagnosis-specific variants (chemo-sensitive) and common diagnosis-relapse variants. The most prominent relapse-specific genetic lesions implicated in chemotherapy resistance in ALL were the gain-of-function mutations in *NT5C2* (22/175, 12%), as previously demonstrated [81,92]. *NT5C2* was found exclusively in relapse samples. Other alterations that were more prevalent in relapsed samples were mutations in the *SETD2* (2.3%), *NR3C1* (1%), *WHSC129* (4.6%), *WT1* (6.2%), and *CREBBP9* (9.7%) genes. *TP53* was observed to be more frequent at diagnosis in adult (4/26) than in pediatric (5/149) cases [93]. A similar study, assessing 103 pediatric ALL triplets (87 BCP and 16 T-cell ALL), observed that relapse-specific somatic alterations were enriched in 12 genes (*NR3C1*, *NR3C2*, *TP53*, *NT5C2*, *FPGS*, *CREBBP*, *MSH2*, *MSH6*, *PMS2*, *WHSC1*, *PRPS1*, and *PRPS2*), all of which were involved in the drug response. The mutational relapse signature had a global prevalence of 17% in the very early relapse group (<9 months after DX), 65% in the early relapse group (9–36 months), and 32% in the late relapse (>36 months) group. The authors examined the mechanism/factors by which the leukemic cell can increase the number of mutations at relapse by analyzing mutational signatures based on the trinucleotide context [94]. A novel mutational signature, thought to arise from thiopurine treatment (6-thioguanine and 6-mercaptopurine), was detected in 27% of relapsed ALLs and was responsible for 46% of the acquired resistance mutations in *NT5C2*, *PRPS1*, *NR3C1*, and *TP53*, [95]. Collectively, a summary of the data explained above highlights the role of specific mutations in *NR3C1*, *TP53*, *NT5C2*, and *CREBBP* in ALL relapse (Figure 3).

As expected, most of the triplets in the two studies mentioned corresponded to BCP-ALL, which limited the possibility of defining relapse mechanisms based on lymphoblastic leukemia type. The work of Oshima et al. is of particular interest for finding no major differences between BCP and T-ALL except for the activating mutations in the *NRAS* and *KRAS* oncogenes, which were primarily early events in T-ALL and more heterogeneously distributed in B-precursor ALL [93]. With the aim of exploring relapse-specific mechanisms in pediatric T-ALL, 313 leukemia-related genes were assessed by targeted deep sequencing (TDS) and multiplex ligation-dependent probe amplification, together with low-coverage WGS, for CNV detections. These techniques detected CNVs in 214 T-ALL patients (67 non-matched relapse and 147 diagnostic cases) and identified activating mutations in *NT5C2* and/or inactivation of *TP53* and/or duplication of chr17 (q11.2; q24.3) in 48% of T-ALL relapse samples. Again, the *NT5C2* was the gene most frequently bearing relapse-specific mutations, although they had no prognostic implications in the pediatric series. Interestingly, *TP53* mutations were very strongly predictive of a second-event relapse [96]. In adult T-ALL cases, data sets analyzing relapse-specific mutations in this subtype of leukemia are very difficult to obtain due to the rarity of the leukemia. However, recently, 19 triplets (DX-remission/germline-RE) from patients treated in two consecutive PETHEMA trials were analyzed by WGS. Data were compared with other open-access genomic data sets of pediatric T-ALL and ALL cases. In this way, clonal mutations in *PHF6* were found to be overrepresented in adult T-ALLs relative to their pediatric counterparts, shared between primary and relapse samples. Alterations in *SMARCA4* (mutations and deletions) were also detected in adult and pediatric T-ALLs, but almost exclusively in relapse malignancies, suggesting a potential role in resistance to treatment [97]. In addition, the authors tried to resolve the outstanding question about relapse mechanisms of whether the treatment (chemotherapy) itself is responsible for genetic variability due to the generation of de novo variants that drive or support relapse. They addressed this by employing computational modeling to estimate the precise divergence time between primary and relapse clonal populations. In the majority of cases, they observed that less than a year passed between the emergence of the relapse clone (single cell) and diagnosis of the primary disease. These findings, and the calculation of the minimum time necessary for the relapse population to reach approximately 7 × 10^11^ cells, the estimated number for a full-grown leukemia, suggested that in most of the 19 adult T-ALL leukemias analyzed, the relapse population was most probably already present before the treatment began, although it was not possible to detect relapse-specific genetic determinants (mutations) at the time of diagnosis [97]. These results are consistent with the observation of a significantly higher level of branched evolution after diagnosis in pediatric ALL cases compared with adult leukemias, suggesting less divergence between diagnostic and relapsed populations in adults [93]. Data supporting this highly heterogeneous branched clonal structure already present at time of diagnosis have also been validated by modeling relapse in immunodeficient mice [98]. However, another study has suggested that treatment may be responsible for the acquisition of new variants that generate resistance, as the proliferating doubling time model used by the authors is not consistent with the model of a pre-existing clone before diagnosis [95]. This is a very controversial point. The inability to detect resistant mutations at the time of diagnosis due, in part, to the sensitivity of the technique used and to the origin of the sample analyzed, in conjunction with the assumptions required to define a leukemia proliferation model, can dramatically influence the resistance model supported [99].

A very recent technique—variant analysis at single-cell resolution—has proved to be the key to detecting rare variants at diagnosis and during treatment. The technique enables the detection of a limited number of variants in a limited number of cells, the sensitivity being boosted by analyzing a larger number of cells [100,101]. Results obtained using this technique in pediatric T-ALL patients have shown that, at single-cell resolution, *NOTCH1* mutations are also highly heterogeneous [102]. The authors promote the utility of the technique in clinics, based on their success in detecting residual leukemic cells at remission time points (two clones at 0.2% and 0.3%, respectively). However, the sensitivity achieved at the single-cell level is far from that already offered by the high-resolution cytometry currently used in clinics (0.001%) (ALL2019, PETHEMA trial), to track residual leukemic cells. Together with that, the variants specifically observed at relapse could not be detected at the time of diagnosis [102], indicating that sequencing at single-cell resolution is not yet capable of identifying mutations causing relapse in patients at the time of diagnosis, or of increasing the sensitivity of tracking resistant clones in patients following treatment, two key issues to solve to apply the technique in a clinical context. 

## 7. Future Perspectives

The amount of genomic information available has grown rapidly in the last decade as the technology developed to support multi-omics data has become more accessible. From the technical point of view, several private services can currently produce omics data at very reasonable prices, and protocols for generating libraries have been optimized and can be performed in a matter of days with a very high degree of reproducibility. Many of the companies involved in omics have developed easily handled software for analyzing data without the need for a specialized bioinformatician. Sequencing costs have also dropped dramatically. The multi-omic information obtained has helped us better understand the natural history of each T-ALL and reveal the oncogenetic processes operating in non-malignant cells. Genetics is also helping us to understand relapse mechanisms and to design new and more specific therapeutic alternatives. However, it is clear that a large quantity of information is still unavailable to us. In this context, the information provided by the non-coding sequences will help us better define the genetic profile of a T-ALL. Exploration of non-coding sequences needs to be accompanied by the improved assessment of variant functionality. Most of the non-coding variants and a large number of coding variants are classified as being of uncertain significance (many in-dels, small insertions, and deletions), so we cannot predict their impact on the disease. In addition, the contribution of small CNVs in the disease are not well stablished. Functional information is also missing. More importantly, in rare cancers such as T-ALL, especially in adult cases, it is difficult to achieve significant genetic redundancy with which to determine what part of all the genomic information is clinically relevant. To overcome these limitations and advance this area of research, genomic data sets need to be assessed in patients included in standardized treatment protocols in which detailed clinico-biological data at diagnosis, during treatment, and at relapse are registered in detail. More broadly still, groups researching ALL/T-ALL should pool their efforts and collaborate to move definitively towards establishing personalized medicine for these rare types of cancer.

## Figures and Tables

**Figure 1 cancers-14-02474-f001:**
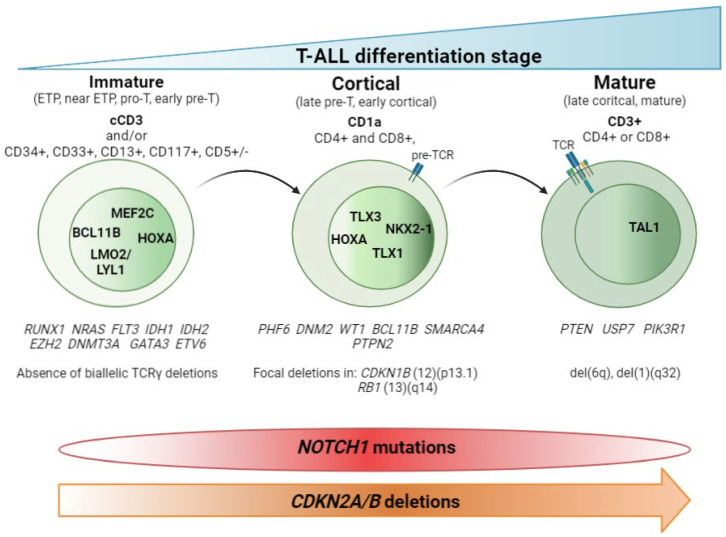
T-ALL classification by stage of differentiation arrest. Schematic representation of the three main T-ALL subtypes according to the blockade of the differentiation process. Hallmark immunomarkers of each subtype are highlighted in bold at the top. Other accompanying immunomarkers, for precise definition of the subtypes, are showed below. TCR maturation is represented in the blasts. Presence of TCR on the blast surface is also a hallmark in the mature subtype. Active transcription factors in each subtype of T-ALL are represented in the nucleus according to the maturation transition. The genes most frequently mutated in each subtype, followed by copy number alterations, are shown beneath. The distributions of *NOTCH1* mutations and *CDKN2A*/B deletions are indicated at the bottom.

**Figure 2 cancers-14-02474-f002:**
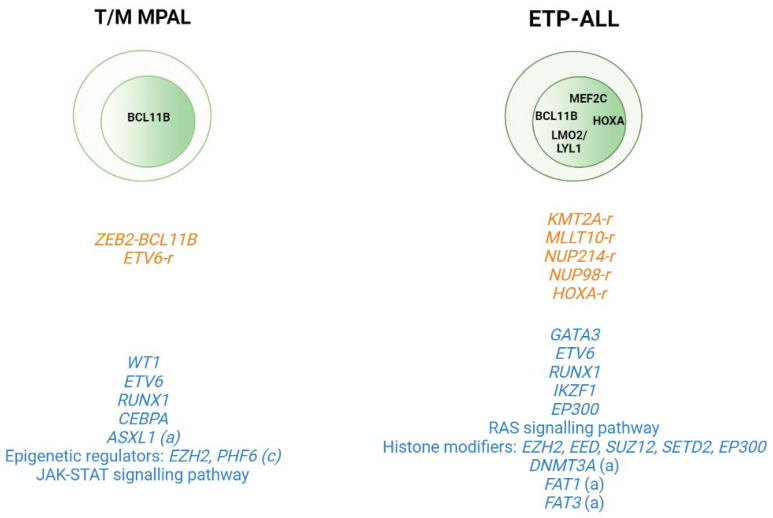
Genomic alterations in T/Myeloid mixed phenotype acute leukemias (T/M MPALs) and ETP-ALL. Active transcription factors in each subtype are represented in the nucleus according to the maturation transition. The rearrangements and fusion genes are written in yellow and most frequently mutations in blue color. (**a**) adult; (**c**) children.

**Figure 3 cancers-14-02474-f003:**
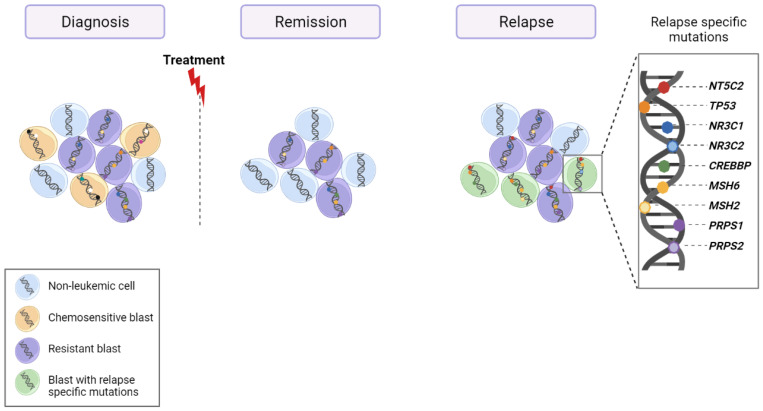
Schematic representation of clonal evolution of leukemic cells from diagnosis to relapse. As described above, relapse blasts contain genetic mutations present at diagnosis and genetic mutations relapse’ specific. The most prevalent relapse-specific mutations are shown at the right side of the picture.

**Table 1 cancers-14-02474-t001:** Non-coding mutations identified so far in T-ALL.

Gene	Affected Region	Variant	Alteration	Functional Impact	Frequency	Reference
*MYC*	1427 kb downstream of *MYC*	Focal duplications	Creation of binding site for *NOTCH1*	*MYC* expression	8/160 (5%) Adult and pediatric	[34]
*TAL1*	8 kb upstream of the transcription start site of *TAL1*	Heterozygous indel (2–18 bp)	Creation of binding motifs for the *MYB* TF	*TAL1* overexpression	8/146 (5.5%) pediatric	[35,36]
*LMO1*	4 kb upstream of the transcriptional start site of *LMO1*	SNV: C → T	Creation of binding motifs for the *MYB* TF	*LMO1* overexpression	4/187 (2.14%) pediatric	[36,37]
*LMO2*	Non-coding region of the exon 2 of *LMO2*	Heterozygous indel	Creation of binding motifs for the *MYB* TF	Activating *LMO2* function	6/160 (3.75%) pediatric 9/163 (5.52%) adult	[38]
*PTEN*	550 kb downstream of transcription start site of *PTEN*	Focal deletions	Deletion of *PTEN* enhancer region	Reduced levels of *PTEN*	5/398 (1.25%)	[39]

Abbreviations: *MYC*: *MYC* proto-oncogene, bHLH transcription factor; *TAL1*: T cell acute lymphocytic leukaemia 1; bp: base pair; *LMO1*: LIM domain only 1; SNV: single nucleotide variant; TF: transcription factor; *LMO2*: LIM domain only 1; *PTEN*: phosphatase and tensin homolog; PE: *PTEN* enhancer.

**Table 2 cancers-14-02474-t002:** Germline variants and predispositions alleles identified in T-ALL.

Gene	Type of Alteration	SNP ID	Alteration	Association with T-ALL: Odds Ratio (P)	Functional Impact	Reference
	Predisposition alterations contributing to the development of T-ALL					
*USP7*	allele	rs74010351	*wt* allele → A Risk allele → G	Discovery cohort → 1.44 (4.51 × 10^−8^)	Downregulation of *USP7* transcription. Risk allele associated with Higher levels of African ancestry and older age at diagnosis	[70]
				Validation cohort → 1.51 (0.04)		
*ATM*	variant		Truncation mutations: R35X 30del215 228delCT	12.9 (0.004)	Aberrant *ATM* protein production → prone to T-ALL	[71]
			Missense mutations: V410A F582L F143C	4.9 (0.03)		
*IKZF1*	variant		N159S	-	Impaired recognition of the target DNA sequences by IKAROS	[72]
*RUNX1*	variant		36171607G > A 36231773 C > T		Risk to develop T-ALL	[73]
		-	p.K117* p.A142fs p.S213fs p.R233fs p.Y287* G365R	-	Loss of transcription factor activity	[74]
	Predisposition alterations affecting drug response and treatment					
*NT5C2*	allele	rs72846714	*wt* allele → A Risk allele → G	-	Higher level of expression of *NT5C2* and lower level of TGN in erythrocytes.	[75,76]
		rs58700372	*wt* allele → T Risk allele → C		Activation of *NT5C2* transcription and reduction of 6-MP metabolism	

Abbreviations: *USP7*: ubiquitin-specific peptidase 7; *wt*: wild type; *ATM*: ataxia-telangiectasia-mutated; *IKZF1*: IKAROS family zinc finger 1; *RUNX1*: RUNX family transcription factor 1; *NT5C2*: 5’-nucleotidase cytosolic II; TGN: thioguanine nucleotides.

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
