# Peer review of "Latest Contributions of Genomics to T-Cell Acute Lymphoblastic Leukemia (T-ALL)"

_cancers, 2022, doi:10.3390/cancers14102474_

Round 1
Reviewer 1 Report
In this review article, Genesca et al report the latest advances in genomics in T acute lymphoblastic leukaemia.
This manuscript is interesting, well organized, but requires amendments before being considered for publication
Main points:
- It should be clearer to the readers that T-ALL is an AYA disease (Adolescent and Young Adult), and T-ALL in old adults represents a small subset with indeed some specific genetic alterations.
-One area of genomics is missing: epigenetics. A small paragraph discussing epigenetics would be beneficial to the manuscript. E.g PRC2 loss; PHF6; the concept of epigenomic translocation with the description of H3K4me3 broad domains that can be created by super-enhancers etc...
-In Chapter 2:
- a) The first sentences are inaccurate. GEa didn’t improve the classification but showed that genomic alterations (mainly ectopic expression of FT by enhancer enhancer hijacking) correlate to specific stages of T-cell maturation. E.g TLX1 ectopic expression blocks rearrangement of TRA (and so blocks the maturation).
There is also a phenotypical classification of T-ALL (EGIL classification).
- b) the NOTCH1-MYC axis, one of the major oncogenic axis in T-ALL, is missing. The description of NMe, a long-range NOTCH-dependent T-cell–specific enhancer, and its oncogenic role, its duplication in about 5% of T-ALL should also be described in table1. (PMID: 25194570)
-In chapter 3:
- a) The first sentence is untrue. T-ALL is an AYA disease. Indeed the peak incidence of ALL is 2-5y but is dominated by B-ALL with ETV6::RUNX1 fusion and High hyperdiploidy standard risk cytogenetics. It has been well demonstrated by Greave’s in monozygotic twins that this is a multi-step oncogenesis with an initiating lesion occurring in utero probably as developmental accident and second hits which lead to Leukaemia. ALL are the paediatric cancer type with the less germline variants described.
Germline variant predisposing to T-ALL are not frequent compared to some solid pediatric tumours (ref 51 is in adult cancer, to add another reference in pediatric cancer would be useful e.g: Genome for kids with 18% cancer predisposing variants but high variation depending of cancer type (PMID: 34301788)
- b) It is important to distinguish GL variants (SNP, polymorphism) leading to a higher susceptibility to develop T-ALL from real GL mutations with a specific phenotype/disease (like ATM, RUNX1-FPD and also Nijmegen breakage syndrome and CMMRD in T-LBL)
RUNX1: to add that familial platelet disorder (FPD-RUNX1) is also associated to T-ALL not only predisposition to myeloid malignancies (PMID: 32208489), other variants are reported.
Table 2 should be revised accordingly and not called germline mutations as it contains also SNP.
- c) pharmacogenomics (polymorphism of genes influencing drug metabolism; here the focus on NT5C2 but there are other genes important) shouldn’t be discussed in a chapter called “predisposition mutations and origin”. This should be removed.
-CH and aging chapter:
- a) in ref 41, the authors couldn’t exclude a GL origin of DNMT3A mutation, and it is the same for ref42. You cannot exclude formally an GL mutation.
- b) ref 42: it should be clearer that it refers to a conference abstract (not a peer-reviewed paper yet), if possible it should to be replaced by a BioRxiv reference.
-relapse chapter:
- a) Mullighan’s paper should be also cited as they have used xenografts to show that relapse may be propagated from ancestral, major, or minor clones at initial diagnosis (PMID: 32793890).
- B) Figure 2 needs improvements, that is not obvious that diag and relapse share alterations and what is exactly the effect of the treatment.
Minor points:
-line 75: CD5 is missing in the definition of ETP-ALL
-Line 77: rarely presents CDKN2A/B deletions and NOTCH1/FBXW7 mutations.
-line 81: cortical subtype is only defined by CD1a expression
-line 83: the reference for NKX2-1 is [22] and not [20]. The sentence is confusing, the expression results from enhancer hijacking.
-Line 85: mature subtype is defined by CD3 and TCR expression on cell surface and no CD1a expression
-Figure 1:
The immunophenotypic description is not totally accurate:
-immature express always cCD3
-cortical: CD1a+ is enough (+/- CD4 CD8)
-mature: TCR+ (no CD1a)
In the arrow: CDKN2A/B deletion (not mutation)
-line 462: CMF and IG/TR allele-specific qPCR (ASO PCR, still gold standard in multiples studies as T-ALL flow MRD is difficult)
Author Response
ANSWER TO THE COMMENTS OF THE REVIEWER:
We thank the reviewer for the overall positive criticisms raised on the manuscript. Their comments and suggestions helped us to improve the manuscript after addressing all criticisms, as detailed below:
Comments to review 1:
It should be clearer to the readers that T-ALL is an AYA disease (Adolescent and Young Adult), and T-ALL in old adults represents a small subset with indeed some specific genetic alterations.
Answer to the specific comment: we have included a sentence in new line 33 to clarify this.
One area of genomics is missing: epigenetics. A small paragraph discussing epigenetics would be beneficial to the manuscript. E.g PRC2 loss; PHF6; the concept of epigenomic translocation with the description of H3K4me3 broad domains that can be created by super-enhancers etc...
Answer to the specific comment: we have included an (epi) genetic section (2.2), in new lines 176-210, to explain the contribution of epigenetic studies in T-ALL.
-In Chapter 2:
- a) The first sentences are inaccurate. GEa didn’t improve the classification but showed that genomic alterations (mainly ectopic expression of FT by enhancer enhancer hijacking) correlate to specific stages of T-cell maturation. E.g TLX1 ectopic expression blocks rearrangement of TRA (and so blocks the maturation).
Answer to the specific comment: we have amended that. New lines 54-56.
- b) the NOTCH1-MYC axis, one of the major oncogenic axis in T-ALL, is missing. The description of NMe, a long-range NOTCH-dependent T-cell–specific enhancer, and its oncogenic role, its duplication in about 5% of T-ALL should also be described in table1. (PMID: 25194570)
Answer to the specific comment: as indicated by the reviewer, we have included a paragraph explaining NOTCH dependent MYC enhancer (new lines 117-125), as well as we have included the new information in Table 1.
In chapter 3:
- a) The first sentence is untrue. T-ALL is an AYA disease. Indeed the peak incidence of ALL is 2-5y but is dominated by B-ALL with ETV6::RUNX1 fusion and High hyperdiploidy standard risk cytogenetics. It has been well demonstrated by Greave’s in monozygotic twins that this is a multi-step oncogenesis with an initiating lesion occurring in utero probably as developmental accident and second hits which lead to Leukaemia. ALL are the paediatric cancer type with the less germline variants described.
Answer to the specific comment: we thank you to the reviewer for his /her comments. We however would like to highlight that first sentence refers to ALL global incidence, been higher in pediatric compared with adult cases, as largely has been demonstrated. We agree that most of the childhood cases will correspond to BCP-ALL, but nonetheless germline information has been explored in this small subset of pediatric T-ALL cases. We try to explain this information in this section. As suggested by the referee, we have re-written some parts of this text (new lines 272-380).
Germline variant predisposing to T-ALL are not frequent compared to some solid pediatric tumours (ref 51 is in adult cancer, to add another reference in pediatric cancer would be useful e.g: Genome for kids with 18% cancer predisposing variants but high variation depending of cancer type (PMID: 34301788)
Answer to the specific comment: we have included the reference, new line 281.
- b) It is important to distinguish GL variants (SNP, polymorphism) leading to a higher susceptibility to develop T-ALL from real GL mutations with a specific phenotype/disease (like ATM, RUNX1-FPD and also Nijmegen breakage syndrome and CMMRD in T-LBL)
Answer to the specific comment: we thank you the reviewer for his/her comment. We have amended that and properly distinguish predisposition alleles vs germline variants, new lines 283-289 and line 300 and 351, in addition to Table 2.
RUNX1: to add that familial platelet disorder (FPD-RUNX1) is also associated to T-ALL not only predisposition to myeloid malignancies (PMID: 32208489), other variants are reported.
Answer to the specific comment: we have included this information in the text (new lines 340-354) and also in table 2
Table 2 should be revised accordingly and not called germline mutations as it contains also SNP.
Answer to the specific comment: we have made changes in Table 2 and incorporate new information.
- c) pharmacogenomics (polymorphism of genes influencing drug metabolism; here the focus on NT5C2 but there are other genes important) shouldn’t be discussed in a chapter called “predisposition mutations and origin”. This should be removed.
Answer to the specific comment: we have organized chapters and titles to be more accurate.
-CH and aging chapter:
- a) in ref 41, the authors couldn’t exclude a GL origin of DNMT3A mutation, and it is the same for ref42. You cannot exclude formally an GL mutation.
Answer to the specific comment: we have re-written the paragraph including the sentence that the referee suggested (new lines 431-438).
- b) ref 42: it should be clearer that it refers to a conference abstract (not a peer-reviewed paper yet), if possible it should to be replaced by a BioRxiv reference.
Answer to the specific comment: we have cited the reference as an abstract (see references section).
-relapse chapter:
- a) Mullighan’s paper should be also cited as they have used xenografts to show that relapse may be propagated from ancestral, major, or minor clones at initial diagnosis (PMID: 32793890).
Answer to the specific comment: we have introduced the citation (new line 459) and we have commented the paper (new lines 518-521).
- B) Figure 2 needs improvements, that is not obvious that diag and relapse share alterations and what is exactly the effect of the treatment.
Answer to the specific comment: we have designed a new Figure 2 (in the new version Figure 3) to show the different mutational scenarios found in relapse and the effect of treatment on that.
Minor points:
-line 75: CD5 is missing in the definition of ETP-ALL
Answer to the specific comment: we have included the marker in the definition (new line 79).
-Line 77: rarely presents CDKN2A/B deletions and NOTCH1/FBXW7 mutations.
Answer to the specific comment: the sentence has been amended in new line 80.
-line 81: cortical subtype is only defined by CD1a expression
Answer to the specific comment: we have renamed CD1 marker (line 84). To clarify to the review that here we mention most frequent immunomarkers in the cortical subtype (characterization), we not talk about immunomarkers that define the group.
-line 83: the reference for NKX2-1 is [22] and not [20]. The sentence is confusing, the expression results from enhancer hijacking.
Answer to the specific comment: we have change reference and modified the sentence (new line 85 and 86)
-Line 85: mature subtype is defined by CD3 and TCR expression on cell surface and no CD1a expression
Answer to the specific comment: here again we mention most often expressed immunomarkers in the mature T-ALL subtypes, not the immunomarkers that define the group.
-Figure 1:
The immunophenotypic description is not totally accurate:
-immature express always cCD3
-cortical: CD1a+ is enough (+/- CD4 CD8)
-mature: TCR+ (no CD1a)
In the arrow: CDKN2A/B deletion (not mutation)
Answer to the specific comment: we have modified Figure 1 according review suggestions.
-line 462: CMF and IG/TR allele-specific qPCR (ASO PCR, still gold standard in multiples studies as T-ALL flow MRD is difficult)
Answer to the specific comment: Here we mentioned experience of PETHEMA group, that uses highs sen

Reviewer 2 Report
In this manuscript Doctors Genesca and Gonzales-Gil provide a well written and exhaustive review exploring the last insights regarding the genomics of T-ALL. They also provide a broad overview on the biological significance of gene mutations.
Comments:
- The authors should proofread the review, as there are a few minor typographical and grammar errors. Please check that the symbols of genes and proteins are respectively in italics and in capital letters.
- The subchapter 2.2 regards T-ALL related immature subtypes. I suggest to contribute with other two little subchapters with reference to cortical and mature subtypes, mostly if you mention them in the previous chapter.
Kind Regards
Author Response
Comments to review 2:
We thank the reviewer for the overall positive criticisms raised on the manuscript. Their comments and suggestions helped us to improve the manuscript after addressing all criticisms, as detailed below:
Comments:
- The authors should proofread the review, as there are a few minor typographical and grammar errors. Please check that the symbols of genes and proteins are respectively in italics and in capital letters.
Answer to the specific comment: we have carefully checked the review and change typographical and grammar errors.
- The subchapter 2.2 regards T-ALL related immature subtypes. I suggest to contribute with other two little subchapters with reference to cortical and mature subtypes, mostly if you mention them in the previous chapter.
Answer to the specific comment: we thank you the review for his/her suggestion. In section 2 in the review, genetic abnormalities in each immunophenotypically subgroup are detailed. However, the new and extended information recently published in immature T-ALL cases, move us to summarize all of them in a separate section. Saying that, inclusion of a cortical and mature subsection would be a repetition of information explained in section 2.

Reviewer 3 Report
This manuscript, review type, written by EG and CGG, with the title of “Latest Contributions of Genomics to T-cell Acute Lymphoblastic Leukemia (T-ALL)” thoroughly summarized the recent advances in the understanding of T-ALL.
T cell ALL/LBL (T-ALL/LBL) is a hematologic malignancy of precursor lymphoid cells committed to the T cell lineage, and accounts for approximately 15 percent of childhood ALL/LBL. An abnormal karyotype is found in 50 to 70 percent of cases of T-ALL/LBL. However, in many cases, rearrangements are not detected by karyotyping but are recognized only by molecular genetic studies. The molecular abnormalities include the TCR rearrangements and non-TCR loci such as KMT2A, NOTCH1, JAK1, and JAK3 mutations, etc.
This manuscript is well written, it is easy to understand and to follow. It is very thorough, it could benefit from a final table summarizing the most features to remember or a section with the “bullet points”.
Minor comments:
- Line 33-35. Regarding the survival rates. These percentages are at 5 years? 10 years?
- Line 44. Regarding “CGHas”. Could you please confirm that the most used acronym is not “aCGH” (Microarray-based Comparative Genomic Hybridization)?
- Line 64. Could you please provide more information regarding “Pre-T1 and Pre-T2/3” subpopulations?
- Line 75. Regarding the early T-cell precursor ALL (ETP-ALL). Could you please confirm that it is defined by the absence of CD4? According to the current WHO 2017 classification, the blasts express cytoplasmic CD3 and may also express CD2 and/or CD4 (page 212).
- Regarding Figure 1. Could you please explain if ETP-ALL is included in this figure?
- The frequency of the non-coding mutations that are shown in Table 1 and discussed in the text between lines 115-169 is very low. How relevant are these mutations for the pathogenesis of T-ALL? How relevant are these genes in the pathogenesis of T-ALL?
- The text between lines 174-221 is very dense, with a lot of information. Is it possible to summarize with a table?
- Line 330. The authors could make a note saying the following: “A driver mutation is an alteration that gives a cancer cell fundamental growth advantage for its neoplastic transformation. It differs from passenger mutations in that these do not necessarily determine the development of the cancer.”
Author Response
Comments to review 3:
We thank the reviewer for the overall positive criticisms raised on the manuscript. Their comments and suggestions helped us to improve the manuscript after addressing all criticisms, as detailed below:
Minor comments:
Line 33-35. Regarding the survival rates. These percentages are at 5 years? 10 years?
Answer to the specific comment: the survival rates are calculated in the UK childhood ALL trials with 15 years of follow-up, as the reference indicate. We do not have included the data in the article, considering that the sentence has full meaning without indicating follow-up of the cohorts.
Line 44. Regarding “CGHas”. Could you please confirm that the most used acronym is not “aCGH” (Microarray-based Comparative Genomic Hybridization)?
Answer to the specific comment: we have checked the manuscript to change aCGH for CGHas.
Line 64. Could you please provide more information regarding “Pre-T1 and Pre-T2/3” subpopulations?
Answer to the specific comment: we have detailed these populations in new lines 65-67.
Line 75. Regarding the early T-cell precursor ALL (ETP-ALL). Could you please confirm that it is defined by the absence of CD4? According to the current WHO 2017 classification, the blasts express cytoplasmic CD3 and may also express CD2 and/or CD4 (page 212).
Answer to the specific comment: we thank you the review for the comment. We have amended the mistake, as CD4 markers is not considered an immunophenotypic markers to define ETP-ALL leukemia, according Coustan-Smith definition, that is actually the same definition that the WHO uses (new lines 74 and 77).
Regarding Figure 1. Could you please explain if ETP-ALL is included in this figure?
Answer to the specific comment: we have explained inclusion of ETP-ALL in the immature subtype in Figure 1 at the figure legend (new line 97).
The frequency of the non-coding mutations that are shown in Table 1 and discussed in the text between lines 115-169 is very low. How relevant are these mutations for the pathogenesis of T-ALL? How relevant are these genes in the pathogenesis of T-ALL?
Answer to the specific comment: we have considered all the information published until now to show involvement of non-coding variants in the pathogenesis of T-ALL. We agree with the reviewer that functional experiments are missed in most of the cases. The NOTCH-MYC enhancer was the only non-coding alteration assessed in mouse models to define its involvement in the pathogenesis of T-ALL. However, as we comment in the review more information will be provided in the near future. With that we will have a clearer view of the importance and significance of these alterations.
The text between lines 174-221 is very dense, with a lot of information. Is it possible to summarize with a table?
Answer to the specific comment: we thank you the reviewer for his/her comment. We have tried to summarize the information provided in this chapter in a new figure number 2.
Line 330. The authors could make a note saying the following: “A driver mutation is an alteration that gives a cancer cell fundamental growth advantage for its neoplastic transformation. It differs from passenger mutations in that these do not necessarily determine the development of the cancer.”
Answer to the specific comment: we thank you the reviewer for his/her comment. We have added the note in new line 405.

Round 2
Reviewer 1 Report
In this review article, Genesca et al report the latest advances in genomics in T acute lymphoblastic leukaemia.
This revised version of the manuscript has improved, however there are a few minor errors:
-Line 79-80: not accurate ETP definition, should be: negative or dim CD5 expression, defined as expression in <75% of the blasts.
-Line 398: to modify this sentence to include the definition (and not a note)
-Typo:
-line 33: “an” instead of “and”
-line 245: e missing in epigenetic
-line 397: have and not are
Author Response
ANSWER TO THE COMMENTS OF THE REVIEWER:
We thank the reviewer for his/her effort to help to improve the manuscript. The comments have been addressed as detailed below (changes can by track in blue color):
-Line 79-80: not accurate ETP definition, should be: negative or dim CD5 expression, defined as expression in <75% of the blasts.
Answer to the specific comment: we have included the definition of CD5 expression as the referee suggest (lines 79 and 80)
-Line 398: to modify this sentence to include the definition (and not a note)
Answer to the specific comment: we have included the definition within the text, as the referee suggest (lines 399 and 400)
-Typo:
-line 33: “an” instead of “and”
Answer to the specific comment: we have substitute and by an
-line 245: e missing in epigenetic
Answer to the specific comment: we have corrected the typo
-line 397: have and not are
Answer to the specific comment: we have corrected the typo
